# The role of civil society organizations (CSOs) in the COVID-19 response across the Global South: A multinational, qualitative study

Adam C. Levine[1]*, Anna Park[1], Anindita Adhikari[2], Maria Carinnes P. Alejandria[3], Benjamin H. Bradlow[4], Maria F. Lopez-Portillo[2], Salma Mutwafy[2], Ieva Zumbyte[2], Patrick Heller[1,2]

1 Watson Institute for International and Public Affairs, Brown University, Providence, Rhode Island, United States of America, 2 Department of Sociology, Brown University, Providence, Rhode Island, United States of America, 3 Department of Sociology and Anthropology, Universiti Brunei Darussalam, Bandar Seri Begawan, Brunei, 4 School of Public and International Affairs and Department of Sociology, Princeton University, Princeton, New Jersey, United States of America

* adam_levine@brown.edu

**Data Availability Statement:** The data that support the findings of this study came from qualitative key informant interviews that cannot be completely

## Abstract

Despite receiving less attention than high-income countries, low- and middle-income countries (LMICs) experienced more than 85% of global excess deaths during the first two years of the COVID-19 pandemic. Due to the unprecedented speed and scale of the COVID-19 pandemic, which placed large demands on government capacity, many LMICs relied on civil society organizations (CSOs) to assist in implementing COVID-19 response programs. Yet few studies have examined the critical role CSOs played in mitigating the effects of the COVID-19 pandemic in low resource settings. This study explored the CSO response to COVID-19 in five of the most heavily impacted LMICs in the Global South. Interviews were conducted from May to August 2021 with a purposive sample of CSO key informants within each of the five countries. A total of 52 CSOs were selected from which 53 key informants were interviewed either via Zoom or by phone. Interviews were coded and analyzed using NVivo or MAXQDA2020. Out of the 52 CSOs selected, 24 were national organizations, 8 were regional, and 20 were local. CSOs fell into six categories: community-based organizations, non-governmental organizations, unions/professional organizations, campaigns/social movements, research organizations/think tanks, and networks/coalitions. CSOs across all five countries adapted their missions, stretched their resources, and performed a wide range of activities that fit into five programmatic areas: food security and livelihood support, public health and medical care, cash transfer programs, risk communication and community education, and needs assessment. This qualitative analysis demonstrates the critical role CSOs played in supplementing government emergency aid response by delivering necessary resources and supporting highly vulnerable populations during the COVID-19 pandemic, as well as the primary challenges they faced in doing so. Given the generally weak state of public capacity in the LMICs studied, this role was vital to responding to the pandemic.

anonymized for public distribution. Information about the data collected, including the qualitative agenda used in the key informant interviews, can be found in S2 Appendix.

**Funding:** This work was supported by a grant from American Jewish World Service (ACL). The funders had no role in study design, data collection and analysis, decision to publish, or preparation of the manuscript.

**Competing interests:** The authors have declared that no competing interests exist.

## Introduction

The World Health Organization (WHO) formally declared COVID-19 a Public Health Emergency of International Concern in January 2020 and a global pandemic just two months later in March 2020. Despite its worldwide prevalence, however, the pandemic did not impact all nations or individuals equally, with wide variability between countries and within countries in both case counts and per capita mortality [1]. While throughout much of the pandemic, media attention focused largely on its impact in the Global North, it was actually the Global South that bore the greatest burden, though lack of access to COVID-19 diagnostics resulted in a significant underreporting of cases and deaths [2]. In fact, while low- and middle-income countries (LMICs) accounted for 4.1 million (69%) of the 5.94 million official COVID-19 deaths reported to WHO between January 1, 2020 and December 31, 2021, they accounted for 15.6 million (86%) of the 18.2 million excess deaths estimated to have resulted from the COVID-19 pandemic during this same period [1]. The ratio of reported COVID-19 deaths to estimated excess mortality ranged widely by region, from 1.43 in high income countries (HICs) to 1.89 in Latin America, 9.46 in South Asia, and 14.2 in sub-Saharan Africa [1]. The massive underreporting of COVID-19 deaths in the Global South provided a pretense for many national governments, international donors, and international non-governmental humanitarian organizations to not actively intervene in LMICs, even as it ravaged local communities and economies. In addition to reduced external support, the invisible nature of the pandemic in these low resource settings also meant that local populations were not well informed or motivated to adopt mitigation measures that could have reduced the heavy toll of the pandemic in their communities.

National and local government response varied widely across countries in the Global South, both in terms of directly addressing the COVID-19 pandemic as well as managing its secondary consequences. In many countries, government sponsored COVID-19 response programs faced significant shortfalls in dealing with the challenges posed by the rapidly changing situation [3, 4]. Gaps in providing essential resources and treating infected patients were further exacerbated in LMICs, where the high burden of cases put increasing pressure on already overstrained health systems [5]. For instance, Mexico, which had low rates of COVID-19 testing and a high prevalence of non-communicable diseases, had one of the highest excess mortality rates recorded worldwide [1, 6, 7].

Due to the unprecedented speed and scale of the COVID-19 pandemic, which placed large demands on government institutions' capacity, and especially on local government institutions that are comparatively weak in the global south, many countries relied on civil society organizations (CSOs) to assist in implementing and monitoring COVID-19 response and recovery programs [5]. Defined as non-profit, voluntary groups organized at the local, national, or international level, CSOs encompass a wide array of organizations including community groups, non-governmental organizations (NGOs), labor unions, social movements, as well as faith-based organizations [8]. From scaling up polio immunization campaigns in South Sudan and Somalia to mitigating and preventing HIV in Nigeria, historically CSOs have had a prominent role in implementing public health initiatives during epidemics and pandemics when local governments were unwilling or unable to do so effectively [9–14].

Throughout the COVID-19 pandemic, unemployment, supply chain shortages, and government imposed stay at home orders (lockdowns) hindered access to even the most basic necessities such as food and medicine, leaving vulnerable populations, such as the elderly, the homeless, children, daily wage workers, migrant workers and refugees, at even higher risk for developing malnutrition or exacerbating pre-existing health conditions [15, 16]. Limited prior research has found that CSOs have played a critical role in providing necessary supplies and

resources when citizens were subjected to strict restrictions during the COVID-19 pandemic. For instance, in Thailand, CSOs distributed food coupons, supplied boxed meals, provided survival bags, and reached out to local businesses to find employment for urban slum residents during the pandemic [17]. Similarly in China, one CSO worked with the civil affairs bureau in the port city of Wenzhou to coordinate the distribution of farm products in rural communities during a pandemic lockdown [18].

Past studies have stressed the importance of investing in CSOs to mitigate the negative impacts of public health emergencies such as epidemics and pandemics and strengthen collaboration with local government [19–24]. However, most published literature on the COVID-19 pandemic to date in LMICs has focused on the effectiveness of government responses, with only two geographically limited studies exploring the role of CSOs [7, 17, 18, 25–27]. Though reviews comparing civil society responses during the COVID-19 pandemic and other public health emergencies have been published, such studies focused primarily on countries in the Global North [19, 28]. The objective of this study was to explore the CSO response to COVID-19 across the most heavily impacted LMICs in the Global South, especially as they related to national and local government responses.

## Methods

### Study setting and design

This qualitative study systematically documents, evaluates, and explains the role that civil society organizations played in the response to the COVID-19 pandemic by examining five separate case studies: Mexico, Kenya, South Africa, India, and Philippines. These five countries were chosen as they were among those most heavily impacted by COVID-19 in each of the following regions of the Global South: Latin America, sub-Saharan Africa, and South/East Asia. In the Sub-Saharan African region, South Africa and Kenya had some of the highest estimated excess deaths in the region, with 302,000 and 171,000 deaths respectively, both of which were more than 30 times higher than the number of deaths reported [1]. India and Mexico had the first and fourth highest number of estimated excess deaths worldwide, with 4.07 million and 798,000 deaths respectively, while the Philippines had the second highest number of excess deaths (184,000) in Southeast Asia [1].

### Researcher background

Six total researchers collected the data for this study, one each for Mexico, Kenya, South Africa, and the Philippines and two for India. Researchers were chosen who had experience living and conducting research within each of the target countries and were fluent in both English as well as at least one of the primary languages spoken within the country. Interviews were conducted in either English or the primary language of the country, depending on the preference of the key informant being interviewed. When interviews were conducted in a language other than English, selected quotes were translated by the researcher into English for inclusion in the manuscript. Researchers were all PhD candidates or graduates in sociology; 5 researchers were female and one was male.

### Data collection

A unified, purposive sampling strategy was used across all five countries to choose a representative sample of CSOs in each setting. First, a mapping exercise was utilized by the research team to document the constellation of civil society actors who have played a key role in the pandemic response within the five selected countries. Through use of web searches,

consultation with local experts, and snowball sampling, a comprehensive listing of CSOs responding to the COVID-19 pandemic was developed for each of the five countries, along with basic information about the geographic reach, populations served, and primary activities of each CSO. Then, the compiled lists of CSOs were narrowed down to a manageable list of roughly 10 CSOs per country while still representing the overall diversity of organizations in each country (S1 Appendix). To ensure diversity in the sample, CSO organizations were selected to reflect variation across the following four dimensions: 1) Geographic scale of operations (i.e. larger national or provincial organizations and smaller community-based groups); 2) Type of organization (i.e. secular, religious, professional/labor, social movements, and consortia/networks; 3) Core activities (i.e. direct service, livelihoods, advocacy, information; and 4) Target population (i.e. general population, specific vulnerable populations).

Once CSOs were identified from each country, key informants were then selected to be interviewed from within each organization. Key informants were generally either the founder, executive director, secretary-general, manager, or in some other major leadership role within the organization. A single qualitative agenda was developed in advance of interviews to guide qualitative data collection across countries (S2 Appendix). Using this qualitative agenda, semi-structured interviews were conducted from May 2021 to August 2021 with key informants from each of the selected CSOs within the five country sites. Interviews were conducted virtually either via Zoom or by telephone and were typically 60 to 90 minutes in length. Interviews were conducted in either English, Spanish, Swahili, Hindi, or Filipino depending on the country. Interviews were recorded and transcribed or detailed notes taken with each respondent's permission. When possible, information obtained from key informant interviews was supplemented by publicly available information on the CSO obtained from its website, media reports, published literature, and other sources.

## Data analysis

A grounded theory (i.e. inductive) approach was used to analyze key informant data from each country. The interview transcripts or notes were coded and analyzed using either NVivo or MAXQDA2020 qualitative data analysis software. Analysis was completed in two rounds. The first round focused on block-coding the interview agenda into separate blocks of text by general topic, while in the second-round, key themes within the master codes were identified. Selected coded segments were translated to English for citation in text. All interview transcripts were coded by the same researcher who conducted the interviews in that country and no double coding was utilized to verify results. Memos were then developed for each country summarizing key themes. A comparative analysis of country level findings was then conducted to identify common themes in CSO activities during the COVID-19 pandemic across these five countries in the Global South. Once these common themes were identified, combined memos, organized by common theme, were written collectively by all authors that incorporated data from each of the five countries.

## Ethics statement

Under the United States' Federal Policy for the Protection of Human Subjects (Common Rule) and Brown University's Institutional Review Board's guidelines, this research was not considered human subjects research as interviews were conducted with key informants who only provided information about their organization and country, not their personal lives or experiences. In addition, no identifiable information about the key informants themselves were recorded or reported.

## Results

A total of 52 CSOs were purposively selected for inclusion in the study based on the four dimensions of organization type, geographic scope, population served, and primary activities (see Table 1 for full list of included CSOs and S1 Appendix for their descriptions). Fig 1 demonstrates the geographic regions represented by these 52 CSOs while Fig 2 demonstrates their geographic scale. The types of CSOs selected were classified into six categories: community-based organizations (CBOs), non-governmental organizations (NGOs), unions/professional societies, campaigns/social movements, research organizations/think tanks, and networks/coalitions. Fig 3 shows the proportion of CSOs falling into each of these six categories.

During the pandemic, CSOs adapted their mission, resources and capacities to deliver necessary services and support to affected populations. Focusing on the gap-filling role of CSOs, this study identified five broad categories of activities in which the CSOs studied were primarily engaged: food security and livelihoods, public health and medical care, cash transfer programs, risk communication and community education, and needs assessment. These categories along with the CSO activities are summarized in Table 2.

### Food security and livelihood support

CSO food delivery services can be primarily split into three types: street feeding, community pantries, and food packages. In Kenya, the local organization called Homeless of Nairobi (HON) expanded its daily street feeding program in the Deep-Sea informal settlement, raising around 20,000 USD from a fundraising campaign to do so. The founder of HON explained that food insecurity increased as a result of disrupted livelihoods caused by pandemic lockdowns.

*Food is our staple. Forget the homeless, even other vulnerable people, jobless people, received food packages; 100 families received food packages that would last them 2 weeks".*

*(CSO leader in Kenya).*

In the Philippines, as the demand for food supply interventions increased, Community Pantry PH (CPPH) started a social movement with the goal of sharing food resources, adopting the slogan "give what you can, take what you need." Touching on the common ethos of bayanihan (community support), this CSO spearheaded the establishment of a total of 6,700 community pantries throughout various parts of Philippines.

*We really experienced being in this lockdown where you know, if you didn't have access to food, and if you didn't have money to basically get deliveries, you would be hungry for two weeks. The reason we're doing the pantry is so that people are fed and that they can stay at home and not go out during this pandemic. Because it's so open, there's a lot of opportunities for people to just step in and help. And it didn't really matter if you know anyone, people didn't know each other. And that's what was the common theme across all pantries.*

*(CSO leader in the Philippines)*

One of the most favored strategies of immediate relief was distributing food packages to communities that faced financial difficulties due to the pandemic. For example, the Seriti Institute in South Africa delivered food packages to informal settlement communities throughout several provinces.

*Our priorities were working with a small-scale farmers and the other was working with community organizations. We played the role as a host for raising money for community*

**Table 1.  Dimensions of representation among sampled CSOs.**

| Organization | Geographic Scale | Type | COVID-19 Efforts | Target Population |
|---|---|---|---|---|
| **Mexico** | | | | |
| Alternativas y Capacidades | National | NGO/Network | Research on impact of COVID on CSOs; Awareness and advocacy | CSO and donors |
| Centro de Derechos Humanos Fray Matías de Córdova | Local | NGO | Relief: health, food; Info workshops; Advocacy: migrant and refugee rights | Migrants and refugees |
| El Caracol | Local | NGO | Relief: food, health, education; Awareness | Homeless people |
| Espacio Migrante | Regional | NGO | Relief: Cash distribution, shelters; Info workshops | Migrants and refugees |
| Frente Nacional de Trabajadores de la Salud (FNTS) | National | Social movement | Protest: physical, social media; Organizing; Advocacy | Healthcare workers |
| Grupo de Información en Reproducción Elegida (GIRE) | National | NGO/Network | Gender Observatory on pandemic impact; Monitoring and Information; Communication | Women |
| JADE Propuestas Sociales y Alternativas al Desarrollo | Local | NGO | Research; Monitoring and Information; Advocacy | Domestic workers |
| Otros Dreams en Acción (ODA) | Local | CBO | Relief: money support, food distribution; Monitoring: economic impact survey; Advocacy: report for Mexico City officials | Deported and returned migrants |
| Tlachinollan—Centro de Derechos Humanos de la Montaña | Local | NGO | Relief: food, shelter; Advocacy | Indigenous peoples, domestic migrants, victims and their families, women |
| **Kenya** | | | | |
| Homeless of Nairobi | Local | Formally registered CBO | Direct provision (food) | Unsheltered youth; Urban informal settlements |
| Kenya Medical Practitioners and Dentists Union | National | Formally registered NGO | Advocacy; Providing PPE and psychosocial support; | Healthcare workers |
| Lifesong | Local | Formally registered CBO | Food provision | At-risk/incarcerated youth |
| MUHURI | National | Formally registered NGO | Advocacy; direct provision (food) | Human rights violations victims; Marginalized communities |
| Safe Hands Kenya | National | Formally registered coalition | Water and sanitation (including masks) | Urban informal settlements |
| Shikilia | National | Formally registered coalition | Cash transfers | Urban informal settlements |
| Touch a Soul | Local | Informal CBO | Direct provision (food) | Urban informal settlements |
| United Sisters of Nairobi | National | Informal CBO | Limited direct provision (food) | Poor communities |
| **South Africa** | | | | |
| Abahlali baseMjondolo (AbM) | Local | Social movement | Food provision; Fighting evictions | Informal settlements |
| C19 People's Coalition | National | Coalition of Organizations | Economic policy activism; Emergency relief; Health provisioning and policy | Variety of vulnerable populations |
| Development Action Group (DAG) | Local | Professional NGO | Emergency relief; Emergency housing and slum upgrading | Informal settlements |
| Health Justice Initiative | National | Professional NGO | Global and local vaccine equity | Variety of vulnerable populations |
| Institute for Economic Justice | National | Professional NGO / Think Tank | Economic policy | Policy-makers |
| Pay The Grants Campaign | National | Informal Campaign | Economic policy | Low-income groups (grant recipients) |
| South African Democratic Teachers Union | National | Trade Union | School safety | Teachers |
| South African Labor and Development Research Unit | National | Research / Think Tank | Economic policy | Policy-makers |
| Seriti Institute | Local | Formal NGO | Emergency relief | Informal settlements |
| Tembelihle Crisis Committee | Local | CBO | Emergency relief | Informal settlement |

*(Continued)*

**Table 1.**  (Continued)

| Organization | Geographic Scale | Type | COVID-19 Efforts | Target Population |
|---|---|---|---|---|
| **India** | | | | |
| Bro.Siga Animation Centre | Local | NGO | Relief; Awareness | Children; informal workers; elderly; transgender people; single mothers |
| Arunodhaya | Local | NGO | Relief; Monitoring & Advocacy; Awareness | Urban poor; children; women |
| Center for Youth and Social Development (CYSD) | Provincial | NGO | Relief; Monitoring & Advocacy | Homeless; urban poor; migrants; women and children; tribal groups; frontline workers |
| Trans Rights Now Collective | Provincial | Social Movement | Relief; Monitoring & Advocacy | Transgender people |
| Hemkunt Foundation | Local | NGO | Relief | General population |
| Information and Resource Center for Deprived Urban Communities (IRCDUC) | Local | NGO | Relief; Monitoring & Advocacy | Homeless; urban poor; migrants; women and children |
| Makaam | National | Network | Relief; Monitoring & Advocacy | Rural women; women farmers |
| Mercy Mission | Provincial | NGO coalition | Relief; Monitoring & Advocacy; Awareness Raising | Homeless; migrants; informal workers; general population; rural areas |
| RSS Pune | Provincial | Religious network | Relief; Awareness Raising | General population; frontline staff; migrants; urban poor; hospital workers |
| Sangtin | Provincial | Union | Relief; Monitoring & Advocacy | Rural women |
| Working People's Charter (WPC) | National | Network / Campaign | Relief; Monitoring & Advocacy | Informal workers |
| **Philippines** | | | | |
| Agro-Eco Philippines (AEP) | National | Social Movement | Human Need; COVID Advocacy; Education | Farmers |
| Angat Buhay/ Kaya Natin Movement (AB-KNM) | National | NGO | Human Rights; Human Need; COVID Advocacy; Public Health; Education | Informal settlers; Youth; Women; Workers; Low-Income groups |
| Assistance and cooperation for community resilience and development (ACCORD) | National | NGO | Human Needs; Advocacy | Low-Income groups; Women; Farmers |
| Coalition of Services of the Elderly (COSE) | National | NGO | Human Rights; Human Need; Advocacy; Public Health; Education | Informal settlers; Elderly; Women; Low-Income groups |
| Community Pantry PH (CPPH) | Local | Social Movement | Human Needs; Covid advocacy | Low-Income groups; Informal Settlers |
| Consortium of Bangsamoro Civil Society (CBCS) | Provincial | Faith-based network | Human Rights; Education | Internally displaced; Indigenous peoples; Youth |
| DAMPA- Damayan ng Maralitang Pilipinong Apo | National | Social Movement | Human Rights; Human Needs; Covid advocacy | Women; Informal settlers; Low-Income groups |
| HIV/AIDS Support House (HASH) | Provincial | NGO | Human Needs; Public Health | Patient Groups; LGBTQIA+ |
| Kythe Foundation Inc. (KFI) | Local | NGO | Human Need; Public Health | Youth; Patient Groups (Cancer) |
| Leyte Center for Development (LCDE) | Provincial | NGO | GBV; Human Needs; Advocacy; Education | Internally displaced; Indigenous peoples; Youth; Women |
| Philippine Rural Reconstruction Movement (PRRM) | National | NGO | Human Rights; Education; Human needs | Low-income groups; Indigenous peoples |
| RH Forum Incorporated (RHFI) | National | NGO / Research Network | Human Needs; Education; Public Health | Patient Groups; Women; Youth |
| Rise Against Hunger Philippines (RAHP) | National | NGO | Human Need; Advocacy | Low-Income groups; Informal settlers |
| Tahanan Outreach Projects and Services (TOPS) | Local | NGO | Education; Livelihoods; Daycare Services | Children; Low-Income groups |

## Regions of the Global South

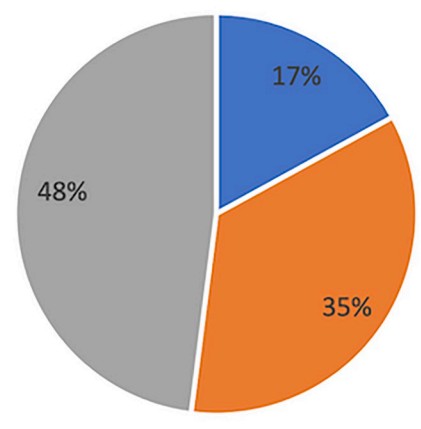

**Fig 1. Geographic region for included civil society organizations.**

*organizations and then to be a partner to the communities themselves. We worked with the small-scale farmers who had fresh produce that they couldn't give to the hotels, restaurants, which were all closed. And they were sitting with huge amounts of fresh produce. So, our model was that we will purchase the fresh produce from all the small scale farmers. We would add the fresh produce to the dry goods, and that would become the food parcels that would then be distributed by the communities themselves. We found we had community organizers who we could pay a stipend and they went out.*

(CSO leader, South Africa)

In Mexico, CSOs also delivered food packages that contained a combination of corn, beans, and rice. One specific organization, Tlachinollan, delivered over 7,000 food packages–each one with 100 kilograms (kg) of corn, 20 kg of beans, 10 kg of rice, 2 liters of oil, and 2 kg of sugar [29]. Similarly in Kenya, a community-based organization called Lifesong provided food packages throughout the duration of the pandemic.

Equally as important as distributing food, several CSOs focused on providing skills and specialized livelihood training services, in order to help the newly unemployed find work. In India, the NGO Bro Siga Animation Center in Chennai conducted online nursing courses, while CYSD in Odisha provided bee keeper trainings for returning migrant workers who had lost employment due to pandemic lockdowns.

## Public health and medical care

In rural areas and in the informal settlements of large cities public health services are often absent or difficult to access due to weak public health infrastructure. Mexico's healthcare infrastructure, for example, has been chronically underfinanced compared to other OECD countries. Total public health spending was 2.8% in 2019. Outside of its major cities, the country has low numbers of medical facilities, and these are poorly equipped and understaffed. A hasty

# Geographic Scales of CSOs

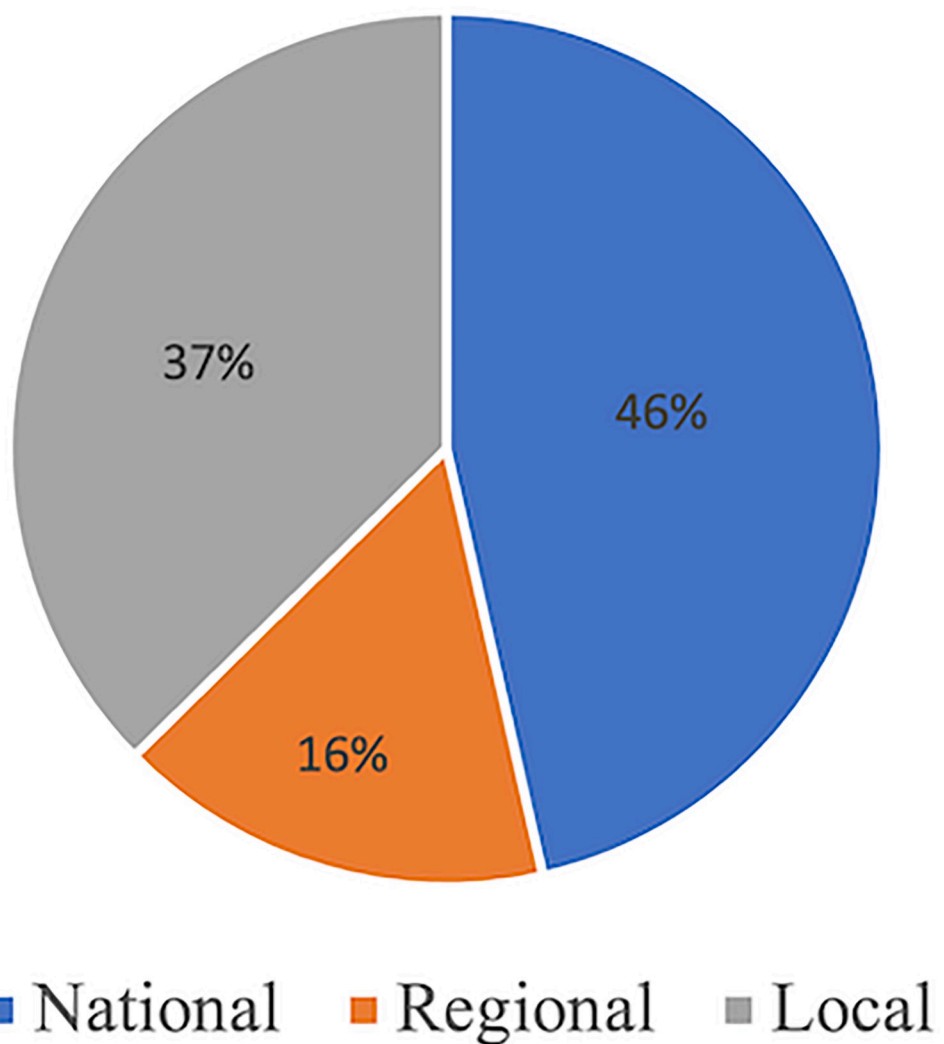

**Fig 2. Geographic scale for included civil society organizations.**

administrative reform to the provision of medical supplies in 2019 led to acute shortages that were felt during the pandemic [30].

Restrictions imposed by government lockdowns during the pandemic further escalated this problem.

*During this pandemic, we were directly involved in a lot of discussions in the policy level side, for example we were still involved in the discussions on reviewing the budget for health vis-à-vis other government budget. How is the budget of reproductive health now especially in provinces? Because pre covid, RH was already in bad situation. Can you imagine the situation now.*

*(CSO Leader in the Philippines)*

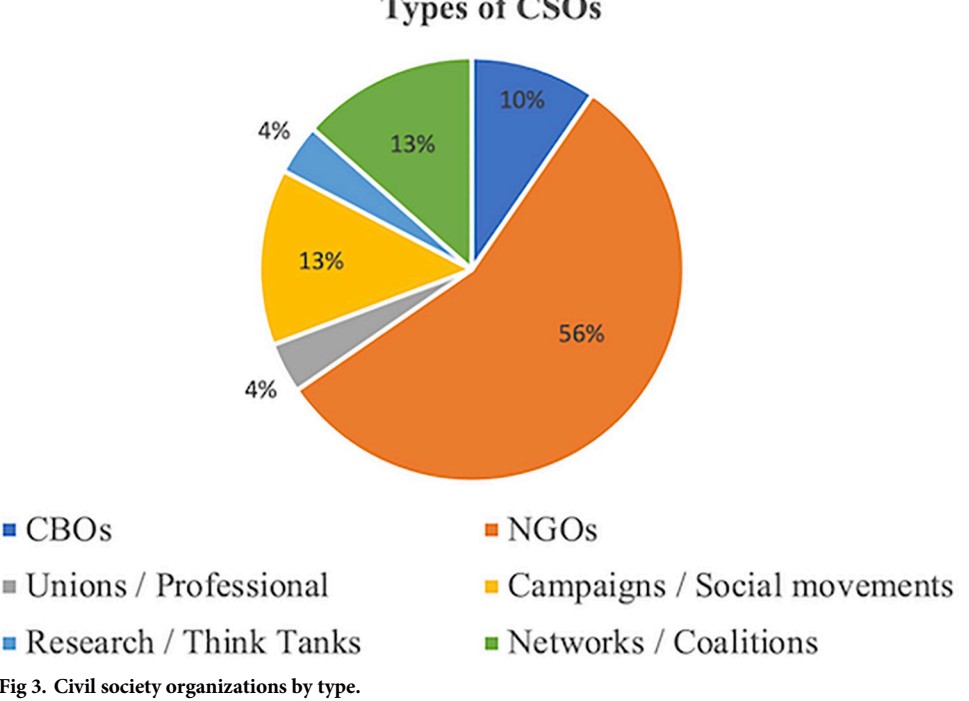

**Fig 3. Civil society organizations by type.**

In many cases, CSOs filled this gap by providing healthcare supplies and well-being services. Such services including supplying hygiene, sanitary kits, and personal protective equipment (PPE), expediting access to oxygen, establishing COVID-19 care centers, providing

**Table 2. CSO activities by major theme.**

| |
| --- |
| Food Security and Livelihoods |
| • Street Feeding |
| • Community pantries |
| • Food packages |
| • Providing skills / specialized training |
| Public Health and Medical Care |
| • Hygiene and sanitary kits |
| • Personal Protective Equipment |
| • Oxygen cylinders |
| • Covid care centers |
| • Consultations |
| • Facilitation of access to specialized health interventions |
| Cash Transfer Programs |
| • Distributing cash transfers |
| Risk Communication and Community Education |
| • Topics such as social distancing, spread, prevention, misinformation, and vaccine hesitancy |
| Needs Assessment |
| • Impacts of the pandemic |
| • Symptom monitoring |
| • Tracking displacement patterns |
| • Conditions of informal settlements |

venues for medical consultations, and facilitating specialized health interventions. For instance, the priority for many CSOs in India was providing people with hygiene kits that contained masks, sanitation products, hand sanitizer, and soap. Similarly in Mexico, CSOs distributed sanitary kits that included antibacterial gels, masks, and valuable information sheets about COVID-19 [31]. In Kenya, a network of CSOs—called Safe Hands Kenya—was formed specifically to provide sanitation stations and raise public awareness about hygiene practices (particularly in informal settlements where it was feared the spread of COVID-19 would be very rapid). Several CSOs in South Africa, India, and Kenya also focused on providing basic PPE and educating the public on how to use and properly dispose of this equipment. CSOs also provided PPE kits to frontline workers and hospitals, shelters, and essential workers, or created public awareness about the use of PPE and social distancing. Emphasizing the importance of their public awareness campaign, a representative from Safe Hands Kenya explained:

> *In the beginning of the pandemic there were myths and misconceptions regarding the virus. The awareness campaign was necessary for the huge impact: give people all the soap in the world, it will make no difference. We tag-teamed with CSOs who are the ones on the ground in communities, each community being different, and saying what needs to be said in different contexts. We had unique approaches that only local CSOs could come up with, like murals on the walls of houses. In informal settlements, walls were real estate and partners on ground asked "why don't you use this?". Especially in appealing to young people, murals were strategically placed in sheng (popular Nairobi slang).*

> *(CSO leader in Kenya).*

To combat the shortage of oxygen in hospitals due to high COVID-19 case rates and hospitalizations, the Hemkunt Foundation distributed large amounts of oxygen cylinders and other medical supplies in Delhi and northern India and, along with another Indian NGO coalition called Mercy Mission, set up oxygen centers. Rashtriya Swayamsevak Sangh (RSS), a Hindu religious association in northern India, also obtained oxygen concentrators and ventilators for hospitals and used public private partnerships (PPP) to establish COVID care centers that provided counseling and other resources to support frontline workers. In the Philippines, some CSOs, such as HIV/AIDS Support House (HASH), aided in providing health services by creating venues for medical consultations, free medicines, and facilitating the access of specialized health interventions.

> *Our major milestone during the pandemic was the provision and delivery of ARV medicines through the help of some private donors at the time that people living with HIV cannot access them due to lockdowns. In partnership with local government units and private sector, we were able to deliver medicines not only in the National Capital Region but also to the other regions where there are shortages of medicines.*

> *(CSO Leader in the Philippines)*

## Cash transfer programs

Several CSOs in Kenya, Mexico, and the Philippines utilized cash transfer programs to distribute direct payments to certain communities. The majority of CSOs in Kenya provided either food security or cash transfer programs, in which CBOs and networks received significant donations from corporate and individual donors. For example, Shikilia, a coalition of private firms and NGOs, provided cash transfers to individuals whose incomes were adversely affected due to the pandemic. The network received donations from global individual and corporate

donors via GiveDirectly and from institutional donors such as Oxfam. Shikilia raised 5 million USD for its cash transfer program, an amount that supported roughly 50,000 households. According to the coalition's website, cash transfer recipients were only enrolled when funds were sufficient to fulfil a monthly commitment of Ksh. 3000 (USD 28 in 2020) for 3 months [32]. Other than the Kenya Medical Practitioners and Dentists Union, all Kenyan CSOs included in this study were involved in the distribution of cash payments to vulnerable individuals and communities. In Mexico, the organization Espacio Migrante reported delivering around 400 electronic debit cards, with funds ranging from 2,000 to 8,000 pesos (approx. 100 to 400 dollars). In the Philippines, some CSOs concentrated their interventions on microfinancing agreements and conditional cash transfers, particularly for households which had lost primary income due to retrenchment or closure of livelihood sources.

> Aside from our information campaign for covid vaccination, we also implemented a cash transfer program for community members whose livelihoods were affected by the lockdowns. We gave out cash support to poorer communities.

> (CSO Leader in the Philippines)

> We operated a food bank project were members can loan money from the organization to finance their small food retail business. Since most of them lost their jobs during the pandemic, this became a helpful alternative for them.

> (CSO Leader in the Philippines)

## Risk communication and community education

Interviewees also reported CSO efforts in raising awareness on topics such as social distancing, virus transmission, prevention, misinformation, and vaccine hesitancy. Awareness campaigns by CSOs, particularly in India, Mexico, and the Philippines, disseminated public health messaging on infection prevention and control measures, medical guidance, and available entitlements. Such awareness campaigns were designed to combat misinformation and vaccine hesitancy. Some awareness campaigns led by CSOs in Mexico and Kenya even provided alternative narratives to government briefings and media reports on the impacts of the pandemic [32]. The Kenya Medical and Practitioner Union (KMPDU) former secretary-general explained that the union started to provide weekly briefings to the press and on their social media pages due to omissions in the Ministry of Health's daily COVID briefings. Key areas in the union's briefings included "a view of the health situation from the bottom" and "data on how member personnel were doing from their active surveillance of doctors getting sick".

In Mexico, a network of organizations released a report in August 2020 on the effects of Covid-19 on migrants and refugees [33]. Another key communication initiative was the Observatory "Gender and Covid-19 in Mexico". Created and coordinated by GIRE in collaboration with 33 CSOs, the Observatory's goal was to track the effects of the pandemic on different subpopulations of women, such as women with disabilities, indigenous and Afro Mexican women, women with HIV, etc. As it grew, the Observatory became a site to collect and diffuse information on the effects of the pandemic on these populations, and the civil society and state responses to them [34].

In the Philippines, the Consortium of Bangsamoro Civil Society helped the government to frame pandemic-related information around Islamic values to increase its acceptance.

> We worked closely with Islamic religious leaders to translate information on COVID-19 preventions including immunization and hygiene to what is acceptable to their followers. We get

*our information from the Municipal Health Office and then have a meeting with Imams so that they could incorporate those in their sermons.*

*(CSO Leader in the Philippines)*

In South Africa, the Tembelihle Crisis Committee (TCC) concentrated on providing basic PPE and education around this equipment by spreading public awareness about the use of PPE and social distancing measures more generally. In India, some CSOs used messaging platforms, like WhatsApp, to promote knowledge of the spread and prevention of COVID-19 and to combat misinformation and vaccine hesitancy. One state level organization in East India, the Center for Youth and Social Development, implemented online citizen support centers to provide awareness on entitlements, medical help, and guidance. Another organization in northern India, RSS, collaborated with social influencers to disseminate information about COVID-19 and with educational institutions to raise awareness about immunity boosting methods at home.

*How do we remove vaccine hesitancy? So that people come forward comfortably and do their vaccines? So they do that. And then we bring out short visual materials, WhatsApp messages. Then we have created a big chain of change agents in the villages where this messages are pushed and then we are in touch with them. If they need any support we can come through this indigenous support center.*

*(CYSD leader, Odisha, India)*

## Needs assessment

In addition to spreading awareness of relevant topics relating to COVID-19, CSOs also highlighted the importance of collecting data for the public good. CSOs would do so by either assessing the impacts of the pandemic on livelihoods, evaluating conditions in informal settlements, monitoring for COVID-related symptoms in populations, or tracking displacement patterns. In South Africa, one of the most useful contributions of The Cape Town-based Development Action Group to the pandemic response was collecting data on the condition of informal settlements in order to inform public policy. It compiled a report on 60 different informal settlements and collected data on each settlement's access to sanitation, clean water, and waste removal services.

*The idea of setting this up was to gather information about hotspots—essentially wherever people are struggling for food, wherever the infection was spreading, where were water and sanitation unavailable. Those kinds of aspects. And really being a conduit to feed this information into other forums where some of this work could be featured and could be utilized in some ways for the response.*

*(CSO leader, South Africa)*

In Mexico, some CSOs created instruments to capture the impact of the pandemic at the community level. For example, Otros Dreams en Acción (ODA), a grassroots organization that provides support for individuals and their families to help them navigate their arrival in Mexico, used a phone survey to evaluate the pandemic's impact on the returned migrants and deportees it serves. Similarly, JADE Propuestas Sociales y Alternativas al Desarrollo, an organization focused on social rights in Yucatan, delivered phone surveys to more than 70 domestic workers to assess how they had been impacted by the pandemic, while the NGO Tlachinollan

monitored symptoms and tracked patterns of displacement and deaths amongst domestic migrants using its shelter over time [33, 35]. By publicly releasing and sharing their findings with other organizations, CSOs such as Tlachinollan and ODA were able to develop recommended priorities for action. Such collaborations allowed for advocacy campaigns to accurately demonstrate the pandemic's impact on specific populations. In Kenya, for instance, one of the key activities of the Kenya Medical Practitioners and Dentists Union was assessing and reporting on how medical personnel were impacted. As mentioned above, the union ran a surveillance system, and in response to the risks facing healthcare workers, applied pressure on the state for better working conditions, and sufficient PPE.

> "*It was a tense moment, in the month of November we lost 4 doctors. I came out guns blazing to the media. We had to use the public to push the agenda, so that on the other side, in the taskforce meetings with the government, we could push things along*"

> (CSO leader in Kenya)

In Chennai, India, the Information and Resource Centre for the Deprived Urban Communities directly met with the municipal authority to inform them about the needs of homeless and children (especially food and shelter) who were overlooked in state's relief efforts. It was the CSO that brought these issues to the attention of the state.

These data gathering interventions played significant roles in filling knowledge gaps and monitoring the pandemic's impacts on especially vulnerable and less visible groups within society. By accurately informing CSO advocacy campaigns and developing evidence-based priorities for action, these efforts often fed into public policy and disseminated valuable data to the government.

## Challenges faced by CSOs

While CSOs across all five countries made significant contributions to pandemic response in the areas detailed above, they also faced many daunting challenges. Two of the key challenges faced by CSOs in the implementation of their pandemic-relevant programs relate to sustainability of funding and personnel security. The need of some CSOs to shift their activities to programs that could support immediate needs (i.e. food and livelihood security) led to the discontinuation of funding from agencies with differing programmatic requirements as well as scaling back or suspending other programs more directly related to their core missions. The director of MUHURI, a human rights advocacy organization in Kenya explained the need, as well as the challenges, in CSOs suspending their core missions:

> *A mission drift was not on my mind. The core values of the institution are the core values of the institution. For MUHURI piecemeal contributions are not our mission, and there is insufficient time for coordinating them, time is a resource. But the dilemma is, you shoot a story about food shortages during the pandemic, how do you not provide it to those you have just covered? It is all these contradictions. And direct food provision got quit a bit of money; resources are often concentrated in a few areas, donors do not speak to each other*".

> (CSO leader in Kenya)

The key informant from the organization Tlachinollan in Mexico, expressed similar concerns:

*"We had to adapt to the pandemic. And it was very tiring, it was having to strategize constantly. We had our strategic projects in January [2020], our strategic planning, and everything changed. And something frustrating was that people and donor agencies would tell us 'Do you want to readjust? and we would say 'Yes, but we don't know how'. Because we didn't have the whole picture."*

(CSO Directive Tlachinollan)

In the Philippines, the securitized approach of the state in governing the COVID-19 emergency put some CSO workers in conflict with armed state actors (i.e. arrest due to lockdown violations while providing support to populations).

*My sister said, 'I was being profiled today. You know, there were five different cops that were there, they kept asking my information throughout the day. They were asking volunteers where I live and all that–" and she didn't realize that it was happening because to her all these are just people who are curious about the operations*

*(CSO Leader in the Philippines).*

CSOs across multiple countries also had to adjust to working online, and initially it was challenging to retrain staff in using various online tools. For many CSOs and especially those that engaged in healthcare related work, there was a lot of learning on the go, as they had to come up with tactics and processes for how to carry out health-related operations that were outside their organization's primary mission. Many organizations also lacked sufficient volunteers on the ground and staff with appropriate skills, such as medically trained personnel and data science specialists.

There were also logistical challenges in delivering relief. During the Delta wave in India when COVID-19 spread was rapid and CSOs were more engaged in direct medical relief than during the initial wave, CSOs had to put more effort in ensuring that volunteers had proper PPE equipment. Beyond that, volunteers who were not used to fieldwork needed insurance and income support. One basic logistical hurdle for many CSOs delivering relief was obtaining permission to travel. As movement was restricted within districts and even cities, CSOs had to obtain travel passes which was a multi-stepped, bureaucratic hurdle that required surveying the populations to establish who would be given aid. In addition to organizational challenges, interviewees reported that operations were challenging for their individual staff, as many workers were also dealing with suffering and death affecting their own families and friends. The Director of the CSO Centro de Derechos Humanos Fray Matías in Mexico expressed this tension:

*"I think that one of the biggest challenges for civil society organizations is that we always focus on acting for other people, but this crossed us completely—I mean we always dealt with the problems of other people, and it's not that we don't see ourselves in them, but those problems do not affect us in the same way. But the pandemic affected us completely, we were afraid of losing a family member, we still are. And that is something we need to recognize, because it forces us to think that we are in a collective and social struggle, but it was very hard for us. (. . .) So I think the emotional and psychological aspect of people working for this organization and others has been very important."*

(Director CHO Fray Matías)

## Discussion

To our knowledge, this is the first multi-country, comparative study that examined, documented, and evaluated the roles that CSOs have played in response to the COVID-19 pandemic in the Global South. As one would have anticipated from the prior literature, CSOs in our study played critical roles in demanding accountability, advocating for marginalized or voiceless communities, and supplementing or filling in where state actors could not or would not act [17, 18]. Our interviews conducted with key informants from 52 CSOs across five LMICs revealed that CSO's primary gap-filling activities during the COVID-19 pandemic fall into five main areas: food security and livelihood support, public health and medical care, cash transfer programs, risk communication and community education, and needs assessment. These findings are in line with the limited prior publications that examined the roles that CSOs have played in other LMICs during the COVID-19 pandemic, though provide more detail on the full range of activities and the challenges faced by CSOs in their implementation [17, 18, 36].

While the need for CSOs to provide support for public health and medical care during a pandemic that often overwhelmed healthcare systems may not be surprising, some of the other activities in which CSOs engaged may be less obvious. For instance, food security and livelihood support were particularly prominent activities for most CSOs across all five countries studied, including providing food to street/pavement dwellers, setting up community pantries, delivering of food packages, and instituting livelihood skills training. Many LMIC governments instituted strict lockdowns at the start of the pandemic, following the lead of high-income countries in the Global North, but without regards to the more significant impacts these lockdowns would have on the livelihoods of their populations [15, 16]. In populations with large proportions of individuals working in the informal sector or for daily wages, lockdowns resulted in rapid food shortages and hunger for millions of individuals, largely by preventing individuals from working and also through disruptions in food supply chains [37–39]. Our research demonstrates that perhaps the greatest contribution of CSOs across the Global South during the COVID-19 pandemic was mitigating the effects of lockdown-related food shortages in vulnerable communities.

While improving food security, supporting medical care, and increasing public health capacity were the most concrete examples of "filling the gap" during the COVID-19 pandemic, CSOs were also involved in many other important but less visible activities. Mirroring a current trend in international humanitarian assistance, national and local CSOs across the five countries examined utilized cash transfers to support vulnerable populations in a rapid, cost-effective and socially distanced manner. In many cases, CSOs were able to utilize their previously developed networks as well as their strong ties to communities in order to both raise and distribute funds at the start of the pandemic, and to conduct rapid needs assessments, which were often context specific. In some cases, these assessments helped inform the activities of CSOs themselves and their larger networks. In other cases, as mentioned above, these needs assessments were shared with government and used for advocacy purposes, in order to improve the efficiency, effectiveness, and accountability of state sponsored aid.

Finally, in an era of misinformation, much of it often spread by governments themselves, CSOs played an important role in getting clear and accurate messaging out to the communities they served. This included public health messaging around the importance of mask wearing, social distancing, and vaccination. It also allowed them to provide accurate information on the impact and significance of the pandemic, sometimes countering government attempts to minimize it.

The role of civil society in supporting the delivery of public goods such as health has previously received a lot of attention in the literature. A review of the literature shows that CSOs in

fact play three distinct roles in enhancing public action [40]. First, CSOs play an important accountability or whistle blower role by holding state and private actors to account. Second, CSOs can play an advocacy role by advocating for those who are marginalized or voiceless. This can include drawing attention to issues or problems that pubic actors have failed to recognize or neglected. Third, CSOs can substitute for or supplement state actions, that is provide services or resources that states can't or won't provide. This gap-filling role can involve basically taking over and providing public goods or extending existing services to populations or areas that the state can't reach. In this paper we focused on the gap-filling role for two reasons. First, the suddenness and severity of the pandemic called for a dramatic scaling up of interventions and to a large degree that scaling up was met through CSOs. Second, the need for CSOs was all the more pronounced in this public health emergency because of existing significant deficits in the capacity of states in the global south.

The pandemic also exposed the problems of state capacity that have been a central preoccupation of the literature on development [41]. While all states faced capacity limitations during the COVID-19 pandemic, LMIC states were much more limited in their basic capacities at the start of the pandemic, including resources and front-line personnel. This is especially true for public health, which is still vastly underfunded in many LMICs [42]. Second, as is true of all states, LMIC states systematically underservice certain populations, including slums, migrant workers, women, ethnic/racial minorities or geographically remote or marginalized areas [43]. Third, local governments in LMICs are generally very weak, with extremely limited resources, personnel and independent decision-making capacity [44]. Given that responses to the pandemic relied heavily on direct interaction with local populations, the lack of strong local institutions and public actors were especially problematic. Overall, then, given these pronounced deficits in state capacity and the ability to mobilize institutional responses in response to the pandemic, we hypothesized that the gap-filling functions of CSOs would be especially important.

Reflecting on these activities we can draw out some larger lessons. First, the gap-filling role played by CSOs in addressing the pandemic confirms what the literature on state capacity has found, namely that there are systemic deficits in state capacity, especially at the local level [42]. CSOs played a key role in providing services and information where state actors could not, but also in reaching populations that were more vulnerable or less likely to be able to make use of existing services. Analytically, we however need to distinguish between two different gap-filling roles. The first is simply providing a service that the state does not. More often than not this is a direct response to problems of state capacity or lack of accountability. The obvious policy lesson is that public health services need to be significantly built up in LMICs. The second is complementing existing state policies or roles. That is, in many of the cases we documented, CSOs actually worked very closely with local officials, engaging in activities that states, regardless of their official capacity, often find more difficult to do. In the literature, this has been referred to as co-production, and specifically forms of intervention that build synergies between the state and civil society [45]. A good example of this is when civil society volunteers work with public health officials to go door-to-door delivering medical supplies or food, something that not even a high-capacity state can do. The policy lesson here is the recognition that while states and CSOs often find themselves in an adversarial position, it is also often the case that they can develop very positive-sum interactions.

This second observation about the gap-filling function in turn raises and important question: what makes CSOs sometimes better at delivering than states? With the important qualification that any answer depends significantly on context, our findings provide 4 tentative responses. 1) CSOs are more flexible than governments: they can mobilize networks and (limited) resources (including volunteers) much more quickly that states generally can. States have

more cumbersome decision-making processes than CSOs and in normal ties this is not a problem and in keeping established administrative practices. But in times of crises, CSO flexibility is critical, though as many of our respondents noted, very difficult to sustain. 2) CSOs can work in the interstices of multi-layered governance and mediate between levels. This is critical, because in any country that has multiple levels of governance (center, state, regional, local), moving decisions and resources between levels runs into political and administrative challenges. In contrast, many CSOs are much more fluid and can work around these choke points. The role that CSOs in Delhi played in delivering oxygen as well as the way in which South Africa's umbrella civil society organizations could coordinate across sectors, are both cases in point. 3) CSOs are capillary in ways many states are not. They reach down into communities and connect to citizens in ways that many states can or do not. As such they can in particular reach within vulnerable or neglected communities. 4) Many CSOs enjoy accumulated trust. To the extent that many grass roots CSOs establish themselves through long years of working with communities, they are much better positioned to mobilize cooperation in times of crisis and to provide information that is seen as reliable.

## Limitations

This study has several limitations. This research relied on interviews with leaders of CSOs, not members of affected populations, so we cannot make inferences about individual or community perceptions regarding the role that CSOs played during the COVID-19 pandemic. While the initial mapping exercise attempted to capture all the CSOs involved in pandemic response across the five countries studied, many smaller CSOs were likely missed. However, the mapping exercise did likely capture the most important dimensions of variability across studied CSOs, ensuring that our purposive sample of roughly 10 CSOs selected per country was largely representative of the full range of CSOs in each country. Interview transcripts were coded using slightly different data analysis software platforms across different countries, though the same qualitative agendas were used to guide the semi-structured interviews and the same grounded theory analysis was used to code the data across all five contexts. All interview transcripts were coded by the same researcher who conducted the interviews and no double coding was utilized to verify results. Finally, this study involves retrospective reports of CSO activities by key informants chosen from their leadership. While these individuals were best positioned to comment on CSO activities during the pandemic, they may have suffered from poor recall or subconscious bias in reporting their own organization's accomplishments.

## Conclusion

Despite the challenges posed by the COVID-19 pandemic worldwide, CSOs across all five countries examined in the Global South adapted their missions and fulfilled a wide range of gap-filling responsibilities that can be primarily categorized in five themes: food security and livelihood support, public health and medical care, cash transfer programs, risk communication and community education, needs assessment. This qualitative analysis demonstrates how innovative and adaptable CSOs are in delivering crucial resources and supporting highly vulnerable populations, illustrating their critical role in public health emergencies. More importantly, this study illustrates how the implementation of CSO interventions can foster human, social, and economic rights, and also supplement state responses to public health challenges. Future research should focus on expanding this research to allow direct comparisons between the roles of CSOs in high, middle, and low-income countries, including the perspectives of affected populations themselves in addition to CSO leaders, and identifying factors that can help and encourage CSOs to provide surge capacity during future pandemics.

## Supporting information

**S1 Appendix. Profiles of sampled civil society organizations.**
(DOCX)

**S2 Appendix. Qualitative agenda for key informant interviews.**
(DOCX)

## Acknowledgments

The authors would like to thank Monique Gainey for her support with editing this manuscript.

## Author Contributions

**Conceptualization:** Adam C. Levine, Patrick Heller.

**Formal analysis:** Anindita Adhikari, Maria Carinnes P. Alejandria, Benjamin H. Bradlow, Maria F. Lopez-Portillo, Salma Mutwafy, Ieva Zumbyte.

**Funding acquisition:** Adam C. Levine.

**Investigation:** Anindita Adhikari, Maria Carinnes P. Alejandria, Benjamin H. Bradlow, Maria F. Lopez-Portillo, Salma Mutwafy, Ieva Zumbyte.

**Writing – original draft:** Adam C. Levine, Anna Park.

**Writing – review & editing:** Adam C. Levine, Anna Park, Anindita Adhikari, Maria Carinnes P. Alejandria, Benjamin H. Bradlow, Maria F. Lopez-Portillo, Salma Mutwafy, Ieva Zumbyte, Patrick Heller.

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
