## [Decision Letter · Decision Letter 0]

10 Apr 2023

PGPH-D-22-02065

The role of civil society organizations (CSOs) in the COVID-19 response across the Global South: A multinational, qualitative study

Dear Dr. Levine,

Thank you for submitting your manuscript to PLOS Global Public Health. After careful consideration, we feel that it has merit but does not fully meet PLOS Global Public Health’s publication criteria as it currently stands. Therefore, we invite you to submit a revised version of the manuscript that addresses the points raised during the review process.

Please follow the COREQ checklist  when you submit the revised version of the manuscript to incorporate the important points.

We look forward to receiving your revised manuscript.

Kind regards,

Nafis Faizi, MD, MPH

Academic Editor

Journal Requirements:

Additional Editor Comments (if provided):

Reviewers' comments:

Reviewer's Responses to Questions

**Comments to the Author**

1. Does this manuscript meet PLOS Global Public Health’s publication criteria? Is the manuscript technically sound, and do the data support the conclusions? The manuscript must describe methodologically and ethically rigorous research with conclusions that are appropriately drawn based on the data presented.

Reviewer #1: Partly

Reviewer #2: Yes

2. Has the statistical analysis been performed appropriately and rigorously?

Reviewer #1: N/A

Reviewer #2: Yes

3. Have the authors made all data underlying the findings in their manuscript fully available (please refer to the Data Availability Statement at the start of the manuscript PDF file)?

Reviewer #1: Yes

Reviewer #2: Yes

4. Is the manuscript presented in an intelligible fashion and written in standard English?

Reviewer #1: Yes

Reviewer #2: Yes

5. Review Comments to the Author

Reviewer #1: Dear Authors,

Thank you for taking the time to prepare and submit this manuscript. The topic is interesting and covers important gaps around how CSOs contributed during the COVID pandemic in LMICs. Below I detail several suggestions. In addition to these points, I suggest the authors look at the COREQ checklist (https://cdn.elsevier.com/promis_misc/ISSM_COREQ_Checklist.pdf) and ensure their manuscript covers all points in the checklist prior to re-submitting the manuscript. One key area for improvement in the article is lack of illustrative quotations in the results section. The discussion section should also be revised, to ensure that the topics discussed and the conclusions drawn relate more directly to the results of the study. I suggest this article be revised and resubmitted. Once revised, this article stands to make an important contribution to the literature.

Methods :

Line 194-195 : The authors say « . Selected coded segments were translated to English for citation in text ». Does this mean that coding and analysis were conducted in the original interview language by a group of researchers, or one researcher who spoke all of the interview languages ? Please specify who within the research team conducted the analysis, and their background. You later mention this in line 551 : « All interview transcripts were coded by the same researcher who 552 conducted the interviews and no double coding was utilized to verify results. » but it should be mentioned during the results, along with the background of the researchers involved in the interviews and analysis, and how the findings across the different sites were ultimately combined.

Results :

Table 1 : Please harmonize language in the « Target Population » column and use people-centered terminology. For example, the phrase « Bottom third of the population » and « indigent » could be replaced with « low-income groups » or similar. « Transgender » could be « Transgender people ». « IP » should be defined or spelled out in full.

Lines 231 through 400 : In the results section I was expecting to see illustrative quotes that demonstrate the points the authors make in the text, as is the convention in most qualitative research. The authors should read through other qualitative research published in PLOS Global Health and similar journals and see how quotes are incorporated, and seek to similarly introduce illustrative quotes.

The assertions in line 260-263 should be supported with citations or quotes from participants, as right now they seem to be speculation by the authors.

Line 301-302 : « Shikilia raised 5 million USD for its cash transfer 302 program, an amount that supported roughly 50,000 households. ». Please specify how long that amount could support 50,000 for.

Line 314 through 319: Consider removing the word « valuable » as this is an opinion of the authors. The idea that certain campaigns « played an essential role » also appears to be the opinion of the authors and is not substantiated. The notion that some campaigns contradicted government messaging should be detailed more, to understand specifically what messages the government was giving and how the CSO messaging differed.

Line 341 through 344 : The intervention was « most useful » according to whom ?

Discussion :

In general, in the discussion section, the authors should take care around the conclusions that can be drawn from information that was gathered only from CSOs and not from the populations the programs sought to benefit. The data collected focuses on the programs implemented by the CSOs and not on the impact felt by the target population. The discussion section is long at at times goes almost an entire paragraph without referring to the direct results of the study. The authors should revise the discussion section, taking care to comment on the results of the study in comparison with existing literature.

The discussion section could benefit from the addition to more citations to back up the assertions made by the authors. For example, at line 423 the authors state « Many LMIC governments instituted strict lockdowns at the start of the 424 pandemic, following the lead of high-income countries in the Global North, but without 425 regards to the more significant impacts these lockdowns would have on the livelihoods 426 of their populations. » Citations should be provide to support this.

Line 473 : Were the problems of state capacity exposed by the pandemic limited to LMICs ? Were these deficiencies not also observed in HICs ? Consider rephrasing to avoid making this look like an LMIC-only problem.

Lines 477 through 483, and 495-498 should all be supported with citations from literature.

The link between discussions of democracy and pandemic preparedness in lines 473 through 493 does not have a substantial tie with the results section. This paragraph could be removed or the link between the results and this discussion point made more clear.

Line 507 : Avoid making generalisations using the word « always ».

Line 540 : One key limitation is the lack of community perspective, as to how well these CSO programs responded to actual community needs. Please mention this in the limitations and consider adding as a topic for future research.

Reviewer #2: Congratulations to the author for a well-written and well-thought-out article. In order to highlight CSO's contribution during the pandemic period, such work should be highlighted.

I think a few minor points should be added to the introduction, highlighting the fact that global south cases are underestimated compared to global north cases. So based on that How does underreporting impact the public health response to COVID-19 in the Global South and the allocation of resources?

2-In the COVID-19 response, what are the differences between the Global South and the Global North in terms of relationships and collaborations between CSOs and other actors (governments, donors, media, communities, etc.)?

3-What are the underlying factors that explain the differences between the Global South and the Global North in terms of the challenges and opportunities they face in the COVID-19 response?

4-How do they ensure adequate representation and diversity of perspectives and experiences? What are the potential biases and limitations of the sampling strategy?

6. PLOS authors have the option to publish the peer review history of their article (what does this mean?). If published, this will include your full peer review and any attached files.

**Do you want your identity to be public for this peer review?** For information about this choice, including consent withdrawal, please see our Privacy Policy.

Reviewer #1: No

Reviewer #2: No

---

## [Editor Report · Decision Letter 1]

8 Aug 2023

The role of civil society organizations (CSOs) in the COVID-19 response across the Global South: A multinational, qualitative study

PGPH-D-22-02065R1

Dear Dr. Levine,

We are pleased to inform you that your manuscript 'The role of civil society organizations (CSOs) in the COVID-19 response across the Global South: A multinational, qualitative study' has been provisionally accepted for publication in PLOS Global Public Health.

Best regards,

Nafis Faizi, MD, MPH

Academic Editor